# Uric Acid and Plant-Based Nutrition

**DOI:** 10.3390/nu11081736

**Published:** 2019-07-26

**Authors:** Boštjan Jakše, Barbara Jakše, Maja Pajek, Jernej Pajek

**Affiliations:** 1Biotechnical Faculty, University of Ljubljana, Jamnikarjeva 101, 1000 Ljubljana, Slovenia; 2Barbara Jakše s.p., 1230 Domžale, Slovenia; 3Faculty of Sport, University of Ljubljana, Gortanova 22, 1000 Ljubljana, Slovenia; 4Department of Nephrology, University Medical Center Ljubljana, Zaloška 2, 1525 Ljubljana, Slovenia

**Keywords:** uric acid, plant-based diets, chronic diseases, purine, hyperuricemia, gout

## Abstract

Plant-based diets (PBDs) are associated with decreased risk of morbidity and mortality associated with important noncommunicable chronic diseases. Similar to animal-based food sources (e.g., meat, fish, and animal visceral organs), some plant-based food sources (e.g., certain soy legume products, sea vegetables, and brassica vegetables) also contain a high purine load. Suboptimally designed PBDs might consequently be associated with increased uric acid levels and gout development. Here, we review the available data on this topic, with a great majority of studies showing reduced risk of hyperuricemia and gout with vegetarian (especially lacto-vegetarian) PBDs. Additionally, type of ingested purines, fiber, vitamin C, and certain lifestyle factors work in concordance to reduce uric acid generation in PBDs. Recent limited data show that even with an exclusive PBD, uric acid concentrations remain in the normal range in short- and long-term dieters. The reasonable consumption of plant foods with a higher purine content as a part of PBDs may therefore be safely tolerated in normouricemic individuals, but additional data is needed in hyperuricemic individuals, especially those with chronic kidney disease.

## 1. Introduction

Plant-based diets (PBDs) are associated with a decreased risk for morbidity and mortality due to most chronic noncommunicable diseases, including cardiovascular diseases, certain types of cancer, metabolic syndrome, type 2 diabetes and obesity [1,2,3,4,5,6,7]. In interventional studies, PBDs were successfully used to prevent and treat severe coronary heart diseases [8,9,10,11], type 2 diabetes [12], early, low grade prostate cancer [13], and obesity [14]. Studies of centenarians across diverse geographical locations consistently associate a PBD pattern with a low incidence and mortality of cancer and cardiovascular diseases [15]. These populations mostly or predominately eat PBDs high in complex and low in refined carbohydrates, and they infrequently consume fish and meat. While this solid and consistent evidence from interventional and observational studies supports the adoption of PBDs for key noncommunicable chronic diseases in developed countries, we observe a common concern between patients and health-care practitioners about the intake of many otherwise health-promoting plant foods (such as legumes, broccoli, spinach) because of their purine content and uric acid (UA) generation potential.

Although the serum UA concentration in each individual represents a complex interplay between nonmodifiable factors, (e.g., genetics), and modifiable factors, (e.g., body weight and lifestyle), the diet and its purine content do play a part in this. In the literature, there is a lack of data evaluating the impact of exclusive PBDs on serum UA, especially from interventional studies, which do not control for this parameter in most cases. In this review, we will focus not only on the impact of some important individual dietary contents and foods but also on the collective impact of various PBD variants on serum UA levels. Purines may originate from body’s endogenous synthesis or be ingested with foods of variable purine content. Since UA is the end product of metabolism of purines, there is a concern about increased serum UA concentrations with the transition from a Western-type mixed diet to an exclusive PBD due to introduction of the regular intake of plant foods with a high purine content, containing more than 200–300 mg purine per 100 g of the product. We will present and summarize the data on serum UA level changes with transition from mixed Western-type diet to exclusive PBDs. Additionally, we review other potentially modifiable diet-related factors that can affect UA levels, such as type of purines ingested and the role of other phytonutrients. Finally, we will describe some modifications of exclusive PBDs that patients with UA-associated health problems could follow.

## 2. Characterization of Plant-Based Diets

The concept of a “plant-based diet” can have various definitions in the scientific literature; from excluding all animal source foods to including “only” a greater intake of vegetables, fruit, fruit juices, cereals, and legumes, while also preserving the intake of fish, pork, and yoghurt. Some scientific publications categorize PBDs by its actual content, e.g., semi-vegetarian (a typical Western-type diet with a reduced frequency of consuming animal source foods), pesco-vegetarian (PBD including seafood with or without eggs and dairy products), and vegan diets (no animal source foods) [16]. Unfortunately, the dichotomous division to vegetarian and nonvegetarian diet does not offer an overall insight into the quality of diet. 

When we talk about health improving dietary interventions and therapeutic effects of the vegan diet, the term “vegan diet” may not be unanimously representative for the beneficial type of a diet. It may include too many highly processed and calorie-dense foods, such as vegetable oils (sunflower, olive, pumpkin oil), exotic fats (coconut and palm fat, cocoa), processed cereals, highly processed packaged food, too much salt and added sugar. All these can adversely change the nutritional composition of the primarily beneficial PBD and, consequently, the effect on the body. The vegan cohort in the EPIC-Oxford study for example, while consuming no animal-based foods, still consumed substantial amounts of saturated fat, and refined carbohydrates [17]. A well-planned vegan diet should include a much larger intake of fruit, vegetables, nuts, and legumes in comparison with other nonvegetarian and vegetarian diets [18]. A “well-planned” refers to the intake of macronutrients and micronutrients that is adjusted according to the individual’s needs [7]. 

Strict and exclusive vegan diet, often referred to as a PBD in a narrow sense, includes fruit, vegetables, wholegrain cereals, legumes, nuts, seeds, herbs, and spices, while it excludes all animal source foods, such as various types of red meat, poultry, fish, eggs, and dairy products [19]. A whole food plant-based diet (WFPBD) is a variant of an exclusive PBD, described by Campbell and Campbell in 2005 [20]. It is based on plant foods that are minimally processed. It includes little or no added fat and excludes refined carbohydrates. WFPBD allows for ad libitum intake of whole grains, fruits, vegetables, and legumes. In moderation, it also includes nuts, seeds, avocados, natural sweeteners, and certain soy or wheat products that do not contain added fat (e.g., tofu). Heavily processed foods, on the other hand, are not included in a WFPB diet. This means avoiding highly refined grain products (e.g., white rice, white flour), foods containing added sugars (e.g., sugar, high fructose corn syrup) or artificial sweeteners, and foods containing added fat or refined fat (e.g., olive oil, coconut and palm oil) [21].

## 3. Uric Acid and Health Outcomes

UA is the end product of purine metabolism, largely derived from endogenous synthesis, but a minor part also arises from exogenous sources such as foods with purine content, alcohol, and fructose drinks. UA is synthesized mainly in the liver and intestines but is also synthesized in other tissues, such as muscles, kidneys, and the vascular endothelium [22,23,24]. Gout is one of the oldest known infirmities caused by hyperuricemia and UA crystal deposition, presenting as inflammatory arthritis with acute onset and later chronic duration [22,25]. Gout used to be described as the disease of kings, but under the contemporary socioeconomic circumstances, it presents with a much higher general population incidence and prevalence than in the past [22]. Estimates of the prevalence of gout in industrialized countries range from 1.4% to 5.2%, and up to 7.6% in the Pacific Islands [26].

UA at normal levels is associated with scavenging free radicals; thus, UA is an important endogenous antioxidant at physiologically appropriate concentrations and is probably needed to reduce oxidative stress in cells of the central nervous system. UA may provide up to 55% of the free-radical scavenging antioxidant capacity in human plasma, making it one of the major antioxidants in humans [27]. The association of UA with health risks is biphasic since low levels of UA are detrimental to neurons, due to impaired antioxidant capacity in the cell [28]. A large cohort study found a U-shaped association between serum UA levels and all-cause mortality; levels between 300 and 410 µmol/L were associated with the lowest mortality [28]. Low serum UA levels are associated with neurodegenerative diseases such as multiple sclerosis, Alzheimer’s disease, Parkinson’s disease, cancer, vascular disease-related dementia [29,30,31,32], and with increased all-cause mortality [33]. A meta-analysis of studies that investigated low serum UA and Parkinson’s disease has shown that patients with the disease had lower serum UA levels than healthy controls [34]. The authors suggested optimizing serum UA levels by supplying adequate purine-rich foods, which might contribute favorably to Parkinson’s disease treatment. 

Hyperuricemia, on the other hand, is defined by the presence of serum UA levels above 404–416 µmol/L [28,35]. Concentrations above 420 μM are regarded as abnormal because that level exceeds the solubility of urate in water; however, urate is more soluble in plasma than in water, and concentrations >600 μM may be tolerated without crystal deposition [36]. Hyperuricemia is a main risk factor for gout but is also associated with chronic kidney disease, cardiovascular diseases, type 2 diabetes, and dyslipidemia [37,38,39,40,41,42,43]. Hyperuricemia is often a consequence of relative renal underexcretion of UA since more than 70% of its excretion is performed by the kidneys [25]. It presents a risk factor for incidence and a more rapid progression of chronic kidney disease, and some studies have shown a reduction in chronic kidney disease progression with UA-lowering treatments [44]. Similarly, UA-lowering treatment has shown promise to reduce blood pressure in patients with hypertension and to improve insulin resistance [36]. Interventional studies addressing all these disease states so far have not yet reached the patient numbers and evidence level to be able to recommend routine UA-lowering therapy for asymptomatic hyperuricemia in hypertension, chronic kidney disease and metabolic syndrome/type 2 diabetes [36,45].

## 4. Dietary Factors and Risk for Hyperuricemia

Purine-free diets do not exist, but foods do vary in their purine content. Previously, it was thought that the avoidance of high-purine foods forms the single basis of an appropriate diet for patients with gout [46]. Studies have shown that purines from plants with a high purine content might potentially increase the risk of UA accumulation, but for the most part, high purine content is found in energy-rich animal meats, fish, and visceral organs such as the liver [47,48]. A recent cross-sectional study from China reported that the “animal products” pattern, characterized by a high intake of fish, fresh meat, animal giblets, and wheat products, was correlated with an increased prevalence of hyperuricemia [49]. Similarly, “animal products and fried food” pattern was associated with asymptomatic hyperuricemia and a negative relationship was found between the "soy legume products and fruit" pattern and asymptomatic hyperuricemia, independent of blood lipids [50]. Other researchers investigated the associations between foods with a high purine content and protein intake with the prevalence of hyperuricemia by using data from a cross-sectional study of 3978 men aged 40–74 years living in Shanghai, China. They found a direct association between seafood consumption and hyperuricemia and an inverse association between the consumption of unprocessed soy legume foods and hyperuricemia [51]. It is necessary to emphasize that not all animal-based foods increase the risk of gout development. Low-fat dairy products were consistently found to be protective. Dairy, calcium, and lactose intakes were evaluated in several studies and were inversely associated with plasma urate [52,53]

A recent meta-analysis of population-based cohorts found that seven foods (beer, liquor, wine, potato, poultry, and soft drinks) and meat (beef, pork, and lamb) were associated with increased serum urate levels, while eight foods were associated with reduced serum urate levels (eggs, peanuts, cold cereal, skim milk, cheese, brown bread, margarine, and noncitrus fruits, [54]. This association was previously reported in two Caucasian cohorts as well [55]. A review of prospective cohort and cross-sectional studies showed that a patient’s individual risk likely represents a complex interplay between nonmodifiable factors, such as age, gender, race, and genetics, and modifiable factors, such as diet, body weight, and lifestyle. However, given current knowledge in the literature, the intake of alcohol, purines from meat and seafood, and fructose- or sugar-sweetened beverages (SSB) has been associated with increased risk of incident gout, whereas dairy products, coffee, vitamin C, and cherries may protect patients from developing hyperuricemia and gout [23]. A meta-analysis that included nineteen prospective cohort or cross-sectional studies concluded that the risk of hyperuricemia and gout is positively correlated with the intake of red meat, seafoods, alcohol, or fructose and negatively correlated with dairy products, soy foods, and fruits (e.g., cherries). Vegetables with high purine content showed no association with hyperuricemia but did show a negative association with gout risk. Coffee intake was negatively associated with gout risk, whereas it may be associated with increased hyperuricemia risk in women but decreased hyperuricemia risk in men [56].

Fructose has gained increased attention as a potential cause of hyperuricemia since its metabolism produces urate as a byproduct [36]. Currently, fructose is a primary source of the sweeteners used in the food supply, as well as a major component of sucrose (table sugar). It is found in SSB, ketchup, various sauces, protein and energy bars and drinks, sweet breads, dairy products, and other formulated food products [57]. One cross-sectional study found an association between plasma UA and SSB, but not with fructose, and the authors suggested that fructose is not the causal agent underlying the SSB-urate association. The authors proposed several reasons for the obtained results but suggested that the relationship between fructose and health outcome is probably dose dependent and might vary due to the administration of fructose in unnatural amounts and forms [53]. In a community-based African American study, researchers found that a dietary intake of fructose and an intake of vitamin C were significantly and oppositely associated with high serum urate concentration and increased risk of hyperuricemia. The magnitude of the association between the fructose:vitamin C intake ratio and both serum urate and hyperuricemia was stronger in men than in women [58]. This association between fructose intake from sugar-sweetened soda and increased risk of incident gout was also confirmed in a large prospective cohort study [59]. 

In addition to sucrose and high-fructose corn syrup, fresh fruits also contain fructose, suggesting that patients with hyperuricemia or gout might also need to avoid fresh fruit. While fruits do contain fructose, not all studies suggest fruits increase the risk for gout. There is a relatively small amount of fructose in an individual fruit and the presence of other nutrients in the fruit (such as fiber, vitamin C, and many important secondary metabolites) may slow fructose absorption or partially block the fructose metabolic effect, UA formation, and inhibit superoxide generation. Also, people who ingest high amounts of fruit are often reducing their intake of refined sugars (such as the sugars in SSB and other products), so their overall fructose intake may be low [60]. This hypothesis was also confirmed in a systematic review and meta-analysis of three prospective cohort studies in which the study results showed an adverse association of SSB and fruit juice intake with the risk of gout, while there was no association with fruit intake [60]. A recent critical review explored the biochemistry of UA production, the relationship between fructose intake and UA production, and how this relationship is linked to cardiometabolic disorders. Researchers evaluated the most relevant discoveries in the field and concluded that it is not yet possible to conclude whether fructose intake alone is the main contributor to increased blood UA concentration [61].

Recently, the purine content in vegetable-based meat substitutes derived from modern protein isolates was analyzed. A group of Czech researchers determined the purine content (adenine, guanine, hypoxanthine, and xanthine) in 39 commercially available meat substitutes and evaluated them in relation to their protein content. The study found, on average, low to moderate purine content per 100 g of tested product, and none of the products exceeded the levels of dietary purines above 150 mg per 100 g serving. The purine content per protein unit was lower than with the three meat products used for comparison in this study (chicken liver, chicken leg, and beef). Choosing these newly developed meat substitutes might therefore represent an acceptable alternative for hyperuricemic individuals [62].

Legumes are an important constituent of PBDs since they represent major protein source of plant-based dieters [63]. Legumes are naturally low in fat, practically free of saturated fat and dietary cholesterol, and have a low glycemic index. In PBDs, legumes provide a significant proportion of fiber and B vitamins, iron, zinc, and other important minerals [64]. The legume food group was found to be an important dietary predictor of longer survival in older people of different ethnicities [65]. Considering UA generation potential, within the legumes food group, especially soybean products deliver moderate and high purine load [47]. In Asian countries, where the high consumption of soy legumes is common, the majority of healthcare professionals believe that soy legume products might increase the risk of developing gout and are contraindicated for gout patients [66]. The macronutrient composition of soy legumes differs markedly from other legumes (e.g., beans, chickpeas, and pea), as the percent of protein calories and the fat content are both higher than in other legumes (approx. 40 versus 3%) and the protein quality is also higher [67,68]. Consequently, soy and soy foods are commonly used by vegetarians through a versatile use in the production of meat analogues and milk substitutes [69]. Contrary to popular belief, a large prospective cohort study (Singapore Chinese Health Study) with 63,257 Chinese adults that comprehensively assessed soy intake suggested that soy and nonsoy legumes are associated with a reduced risk of gout [70]. Soy protein may increase serum UA; however, the expected increase in response to amounts typically ingested by Asian people would almost certainly be clinically irrelevant [66]. In general, diets rich in plant foods were not found to be associated with an increased risk of hyperuricemia and gout, even when plant foods with high purine content such as soy legume products were consumed [48,51,71,72]. The British Society for Rheumatology, in their recommendations for the management of gout, recognized these recent findings and stated that inclusion of soybeans and vegetable sources of protein in the diet should be encouraged, but high-purine foods at the same time avoided [73]. High-purine vegetable protein sources are defined below (See Table 2).

## 5. PBD Variants and Serum UA Concentration

PBDs are typically based on the consumption of grains, legumes, vegetables, fruits, and nuts; however, there are many PBD variants with different UA generation potential. PBDs may exclusively contain only plant food sources, while lacto-ovo-vegetarian diets include dairy and/or egg products [7]. Other plant-rich diets, such as The DASH (dietary approaches to stop hypertension) or Mediterranean diet, also include, but to a lesser extent than Western-type diets, other meat-based products [74,75]. The differentiation between the effects of exclusive PBDs and vegetarian diets on serum UA levels is especially important, since it is known that the consumption of dairy products in vegetarian diets decreases serum UA levels and exclusive PBDs may replace dairy products with larger quantities of high-purine plant food sources.

Regarding vegetarian diets, a small short-term study was completed in Germany in which researchers introduced ten healthy male participants to a self-selected meat-containing diet for the first two weeks. After this period, the researchers introduced the participants to three different standardized diets for a period of 5 days each, namely, a Western-type diet that was representative of typical dietary habits, the balanced omnivorous diet, and the ovo-lacto-vegetarian diet. The risk of UA crystallization was highest during the ingestion of the self-selected meat diet and Western-type diet, but the ingestion of the vegetarian diet led to a significant reduction in the risk of UA crystallization by 93% compared to the Western-type diet [76]. Additionally, the results from two prospective cohort studies associated vegetarian diets with lower serum UA and a lower risk for hyperuricemia and of gout. Due to the small number of vegans who eventually developed gout in both cohorts, the analysis of the effect of a vegan diet was not possible, and vegans were classified as vegetarians in the main analysis [48]. This classification is problematic for translating this result from participants who are mostly vegetarians to people who consume exclusive PBDs, since it is recognized that the consumption of dairy products by lacto-vegetarians is clearly linked to lower serum UA values [23]. Results of three representative studies that compared UA serum concentrations in vegetarians and non-vegetarians confirm generally lower UA values in vegetarian samples (see Table 1). 

The DASH diet is a moderate form of a plant-rich diet and is described as being similar to “lacto-vegetarian” diet. The DASH diet emphasizes fruits, vegetables, low-fat dairy products, reduced meat consumption (reductions in saturated and total fat and cholesterol) and the selection of fish, chicken, and lean meat. The diet goals were designed to create a dietary pattern that would have the blood pressure-lowering benefits of a vegetarian diet, yet contain enough animal products to make it palatable and acceptable to the general nonvegetarian public [74]. Researchers prospectively examined the relationship of the DASH “plant-rich” diet and Western-type diets with the risk of gout in 44,444 men without a history of gout at baseline. The DASH diet was found to be associated with a lower risk of gout than the Western-type diet, which was found to be associated with a high risk of gout. The efficiency of the DASH diet was further confirmed in several interventional studies, including in participants with hyperuricemia and in individuals with hypertension with high baseline serum UA levels [79,80]. 

Considering exclusive PBDs, researchers performed a cross-sectional analysis of the EPIC-Oxford Cohort where they compared serum UA concentrations in meat eaters, fish eaters, vegetarians, and vegans [71]. The researchers found that male individuals who exclusively consumed a PBD had higher serum concentrations of UA (340 µmol/L) than meat eaters (315 µmol/L), fish eaters (309 µmol/L), and vegetarians (303 µmol/L). In women, serum UA concentrations were also slightly higher in vegans (241 µmol/L) than in meat eaters (237 µmol/L), vegetarians (230 µmol/L) and fish eaters (227 µmol/L). However, to objectively assess the risk of UA elevation with exclusive PBDs, we would need data from interventional studies that would measure the association between exclusively PBDs and serum UA status. For causality, we need to have well-designed long-term interventional studies, in which people’s diets are changed, and their serum UA status is measured. 

We performed a single armed two-phase interventional study on a supplemented WFPBD looking into cardiovascular risk factors and we also assessed serum UA concentration. The baseline serum UA concentrations were significantly higher in the male participants than in the female participants (387 ± 77 vs. 287 ± 57 µmol/L). Baseline preintervention diet of 36 participants was a typical Western-type diet: most meals were composed of some animal-based food (cow’s milk, yogurt, cheese, cottage cheese, meat from various sources, eggs, and fish) and refined wheat flour-based food (bread, pasta, and pastry). Drinks included SSB and fruit juices. Food preparation contained usage of various vegetable oils and fats. Whole (unrefined) plant foods were largely absent from most meals, and a minority of meals included significant portions of fruit and vegetables in whole food form. The interventional PBD was then largely composed of food groups with moderate and high purine content such as nonsoy and soy legumes, nuts, seeds, brassica vegetables, spinach, mushrooms on weekly basis and a limited amount of fructose found in an unrefined form in fruits and in a refined form in two meal replacements (up to 10 g total). In phase 1 (after 10 weeks of PBD intervention), UA increased by 26 and 10 µmol/L in males and females, respectively. The 95% confidence interval for a rise in UA serum concentration was 1–28 µmol/L (*p* < 0.01) in the whole sample. In phase 2 (after 36 weeks of PBD), UA changed by 1.5 and −14 µmol/L in males and females, respectively. In the subsample that completed both study phases, the final UA content did not significantly differ from the baseline value (321 µmol/L vs. baseline 309 µmol/L) [72]. Additionally, preliminary data from our observational study (https://clinicaltrials.gov/ct2/show/NCT03976479) in long-term WFPB dieters (average duration of PBD diet 5.4 years, no-purine intake limitation), has shown that the average UA serum concentration in males (*n* = 12) and females (*n* = 18) was 328 ± 41 and 246 ± 42 µmol/L, respectively. There was no subject with serum UA concentration above the upper limit of normal range, and the maximum value of serum UA concentration was 380 µmol/L (unpublished observation). In fact, the average concentration in females (246 µmol/L) was below the above cited optimal concentration range 300–410 µmol/L associated with the lowest mortality. Thus, even somewhat larger intake of purine rich plant-based foods, especially in females, could have been tolerated in this cohort.

The results show that the transition from a Western-type to an exclusive PBD containing plant-derived foods with moderate and even high purine content in subjects with normal UA values may be associated with a minimal increase of serum UA concentration well within the normal range. We found no signal that maintaining long-term exclusive PBD would lead to hyperuricemia, even with a diet including high-purine content plant foods. On the contrary, the average concentration in males on a long-term exclusive PBD was at the lower margin of a putative optimal concentration range and below it in females. However, we would need additional prospective data in patients with pre-existing elevated UA serum concentrations, especially in populations with chronic kidney disease and/or gout. So far, the high intake of low-purine foods, such as dairy products, cereals, nonsoy legumes and some form of soy products (e.g., green soybean, bean curd lees (Okara), deep-fried tofu), vegetables, and most mushrooms to manage gout, hyperuricemia, and cardiovascular disease is strongly recommended [47]. Concerning gout, a large prospective study over a 12-year follow-up period of 47,150 men without a history of gout at baseline examined the relationship between purported dietary risk factors and new cases of gout. These researchers found no association with the consumption of plant foods with a relatively higher purine content such as peas, beans, lentils, spinach, mushrooms, and cauliflower. The multivariate relative risk of developing gout among the men in the highest quintile of vegetable-protein intake showed a 27% lower risk than those in the lowest quintile [52]. 

## 6. Purine Content of PBD Lifestyle May Not Be an Unavoidable Challenge

Purines naturally occur in all plant foods. It was found that at least 10–15 mg of purine per 100 g of food is present in all plant foods, with most plant foods containing low or moderate concentrations [81]. The purine content in animal sources (e.g., meat and fish) generally varies from approximately 120 to over 400 mg of purine per 100 g, while the purine content in plant sources (e.g., most nonsoy legumes, grains, seeds, fruits, and other vegetables) varies from 7 to 70 mg of purine per 100 g [47]. However, some plant foods may contain higher purine concentrations, such as 100–500 mg of purines per 100 g food [47,81]. Purine content of some selected plant-based foods is represented in Table 2.

Japanese Guidelines for the Management of Hyperuricemia and Gout recommend the daily intake of dietary purines to be less than 400 mg to prevent hyperuricemia and gout development. Fructose, alcohol and energy restriction, appropriate water intake, and physical activity have been added to these recommendations [83].

Since some types of plant-based foods contain a relatively high purine content, this may lead to a concern that a transition from a Western-type diet to a heart-healthy PBD would not beneficially affect UA levels or could even worsen them. A typical example of such a clinical scenario is a patient with chronic kidney disease grade 3–5 with hypertensive nephro-angiosclerosis where we typically see elevated UA serum concentration. Adoption of a PBD could be very much beneficial for such a patient; however, a concern about the possible aggravation of UA levels should be addressed. Also, due to a larger potassium food content, serum potassium levels need to be monitored as well—we recommend additional control at seven and then 21 days after transition to a PBD in chronic kidney disease patients grade 3b, 4, and 5. In Table 3, we propose dietary plan solutions with such a transition. The key is to replace plant foods of a high purine content with foods of a low to moderate purine content. Examples of diet modifications included replacements such as: soy–nonsoy legumes, broccoli sprout–cauliflower, spinach–red beet, orange juice–hibiscus tea, soy tofu–chickpea hummus, dried shitake–raw or steamed shitake (see Table 3).

The absolute intake of soy legume products needs to be considered in a meal plan design, since people in Western countries increasingly adopt PBDs and soy legume products often represent a major protein source. Soy legume intake varies among cultures, but some studies of vegetarians and vegans in Japan, Europe and the United States reported legume intake to be between 8.4 and 30 g per day [68,84,85]. Researchers from EPIC-Oxford Cohort reported that in female vegans an intake was up to 30 g per day of soy protein [69] which might translate into approximately 100 g of soybean intake per day (low purine content) or 350 g of soy tofu per day (very high purine content). With such quantities, a proper choice of the type legume with a smaller purine content is important (see Table 2 and Table 3).

There are additional factors besides purine content of PBD, which determine the individual UA level response to PBD transition. First, Messina et al. suggest that approximately two-thirds of the total amount of urate in the body is endogenously synthesized and that “only” one-third comes from dietary purines [66]. The production of serum UA depends on the balance between purine ingestion, de novo synthesis in cells, recycling, and the degradation function of xanthine oxidase at the distal end of the purine pathway [86]. The association between a purine-rich diet, increased plasma UA, and gout has long been recognized but a purine-rich diet may be responsible for only 59–118 µmol/L of serum UA [86]. The impact of dietary purines on plasma UA concentration is further modified by the frequency of consumption, cooking methods, bioavailability, types of purines, the presence of other plant-based food components, and lifestyle factors [47,53]. Thus, dietary purine content alone does not represent the major impact on serum UA concentration.

Second, new discoveries in this field highlight the importance of the type of purines present in foods, particularly adenine and hypoxanthine content [47,87,88]. Previously, it was assumed that there were no differences in the uricogenic effects of individual purines (i.e., adenine, guanine, hypoxanthine, and xanthine), but now it is known that among the purines, adenine and hypoxanthine are considered far more uricogenic than guanine and xanthine [87]. More than 60% of the total purines in all cereals, beans, soy legume products, seaweeds, dairy products, mushrooms, and vegetables are adenine and guanine, while >50% of the total purines in most animal and fish meats are hypoxanthine [47]. Additionally, processing methods, such as stewing, roasting, boiling and frying, can decrease the purine content of foods [88]. Concerning the type of purines present in the foods, serum UA is produced by the conversion of the purine derivative hypoxanthine to xanthine and xanthine to UA [66]. High-purine plant-based food has a lower proportion of hypoxanthine purines than animal-based purine sources in meats, fish and some shrimp, and hypoxantine effect on UA generation seems stronger [47,88].

Third, PBDs include various protective components that at least partially contribute to lowering of serum UA levels. Fruit, vegetables, and various herbs are composed of many phytochemicals, and other micronutrient compounds that have been shown to inhibit UA synthesis and were implicated as alternative or additive drugs for gout [81]. Besides cherries and coffee already mentioned above [23,89], several other plant foods such as Indian and Chinese medicinal plants [90,91] and an overall more alkaline diet rich in vegetables and fruit [92] were shown to help in decreasing UA level. Finally, lifestyle factors that often associate with a PBD choice are a lower consumption of alcohol and a lower intake of high fructose-rich beverages which may also help to reduce the UA generation [48].

## 7. Conclusions

The conventional approach to gout prevention and treatment employs a diet low in purine and total protein in addition to limited alcohol intake and weight loss. Such a diet, however, may offer limited efficacy and poor satiety control if the decrement of animal-based protein would promote increased consumption of refined carbohydrates (including refined fructose) and unhealthy fats (i.e., saturated and trans fats) that can worsen cardiovascular outcome [93]. PBDs are predominantly centered around whole grains, legumes, fruits, and vegetables and these foods must be considered as a part of a healthy diet. Concerning the UA generation potential of the diet, the elimination of these foods may be unnecessary and potentially harmful to the health of people since these foods provide many nutritional benefits. Despite some outdated guidelines that continue to promote the avoidance of high-purine vegetables, it is important to emphasize that there are no data from long-term cross-sectional or interventional studies that would show that high-purine plant-based foods represent a clinically meaningful increased risk for hyperuricemia or gout development [52,53]. Various reviews cited above confirmed that high-purine vegetables (in the contrast to high-purine foods of animal origin) showed no apparent association with hyperuricemia or gout development but rather a negative association with gout development risk. 

As regards the relationship between the cumulative impact of plant-based nutrition and serum UA level, studies that compared UA serum concentrations in vegetarians and nonvegetarians have consistently shown a lower mean UA serum concentration in vegetarians. The concern about increased serum UA concentrations in the transition from a Western-type mixed diet to an exclusive PBD from whole food sources, with the regular intake of high-purine PB foods, is not needed in healthy normouricemic individuals since our results show that the increase is minimal and well within the normal range. Especially females on such a long-term diet could even further increase the intake of purine rich plant foods to be able to reach a putative optimal serum UA concentration range. On the other hand, high-purine plant-based foods should be eaten with caution or replaced with lower purine content alternatives if a patient already has advanced kidney disease or gout. In these patients, UA lowering pharmacological agents remain the mainstay of therapeutic approach. We need more data about the impact of exclusive PBD in this patient population; however, lower purine content plant foods are readily available and may be included in a healthy and nutritionally adequate plant-based nutrition plan, an example of which we have presented in this review. 

## Figures and Tables

**Table 1 nutrients-11-01736-t001:** Comparison of serum UA concentrations in vegetarians and nonvegetarians in cross-sectional studies.

Reference	*N*	Type of PBD	UA Serum Concentration (µmol/L)			
			Vegetarians		Non-vegetarians	
Szeto et al. (2004) [77]	30	vegetarians	239		306	
Yang et al. (2011) [78]	171	vegetarians	343		357	
Haldar et al. 2007) [56]	31	vegetarians (6 vegans)	283		291	
Schmidt et al. (2013) [71]	844	Vegetarian & vegan	Vegetarians		Vegans	
			Males	Females	Males	Females
			303	230	340	241

Legend: UA, uric acid; PBD, plant-based diet.

**Table 2 nutrients-11-01736-t002:** Typical purine content of representative plant-based foods (adopted from Kaneko et al, 2014; Davis, 2015 [47,82]).

Plant-Based Food	Purine Content: Amount Per 100g				
	Very High (>300mg/100g)	High (200–300mg/100g)	Moderate (100–200mg/100g)	Low (50–100mg/100g)	Very Low (< 50mg/100g)
Shitake (dried)	**x**				
Nori seaweed (dried)	**x**				
Soy tofu (freeze dried)		**x**			
Hijiki seaweed (dried)		**x**			
Wakame seaweed (dried)		**x**			
Parsley		**x**			
Shitake (for broth, dried)		**x**			
Jew’s-ear (dried)			**x**		
Spinach (young leaf)			**x**		
Fermented soybean (Natto)			**x**		
Broccoli sprout			**x**		
Broccoli				**x**	
Cauliflower				**x**	
Deep-fried tofu				**x**	
Azuki bean (dried)				**x**	
Board bean				**x**	
Red bean, cooked				**x**	
Lentil, cooked				**x**	
Oats, dry				**x**	
Flaxseed				**x**	
Spinach (leaf)				**x**	
Green peas (canned)					**x**
Garbanzo beans, cooked					**x**
Peanut					**x**
Walnuts					**x**
Almonds					**x**

Legend: x, denotes the categorization of food under the respective appropriate purine content.

**Table 3 nutrients-11-01736-t003:** Modification of a whole food plant-based diet (WFPBD) meal plan to lower purine content.

Meal Plan	WFPBD (Moderate to High Purine mp)	WFPBD Adjusted (Low Purine MP)
Breakfast	Oat meal in soya milk, blueberries, and raisins; green fruit smoothie (carrot, young leafy spinach, broccoli, dates, apple, banana, orange)	Oat meal in nonsoy plant-based milk (e.g., oat, almond, rice), blueberries, and raisins; green fruit smoothie (carrot, red beet, kale, dates, apple, banana, kiwi)
Morning snack	Millet with seasonal fruits, hazelnuts, orange juice, whole grain bread with peanut butter	Millet with seasonal fruits, hazelnuts, hibiscus tea, whole grain bread with peanut butter
Lunch	Bean/lentil vegetables soup with parsley; whole grain pasta with tomato sauce, corn and peas, mashed (sweet) potato with soya milk; steamed broccoli; green and tomato salad with pumpkin seeds and balsamic vinegar; red wine	Bean/lentil vegetables soup with minimal amount of parsley; whole grain pasta with tomato sauce, corn and peas, mashed (sweet) potato with nonsoy milk, steamed cauliflower; green and tomato salad with pumpkin seeds and balsamic vinegar; lemonade
Afternoon snack	Sandwich (whole grain bread, non-smoked soya tofu, tomato, red onion, kale or red cabbage); cup of cherries/berries	Sandwich (whole grain bread, chickpeas humus paste/avocado, tomato, red onion, kale or red cabbage); cup of cherries/berries
Dinner	Buckwheat porridge with dried shitake mushroom; mixed green salad: green leafy vegetables, boiled potato, tomato, walnuts/shredded almonds; figs	Buckwheat porridge with steamed or roasted shitake mushroom or risotto with brown rice, corn, tomato, mixed green salad: green leafy vegetables, boiled potato, tomato, walnuts/shredded almonds; figs

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
