# Peer review of "Uric Acid and Plant-Based Nutrition"

_nutrients, 2019, doi:10.3390/nu11081736_

Round 1

Reviewer 1 Report

Uric acid and plant-based nutrition evaluation:

animal-based food sources (e.g., meat, fish and animal visceral organs), some plant-based food sources (e.g., certain soy legume products, sea vegetables and brassica vegetables) also contain a high purine load.

Authors conclude that reasonable consumption of plant-based (PB) foods with a higher purine content as a part of PB Diets (PBDs) may, therefore, be safely tolerated in normouricemic individuals, but additional data is needed in hyperuricemic individuals, especially those with chronic kidney disease.

Comments

1. This is a communication paper. on an interesting issue suggesting that the conventional approach to gout prevention and treatment employs a diet low in purine and total protein in addition to limited alcohol intake and weight loss. 

.

2. The text is clear and informative

3. References are up to date.

Author Response

We are thankful for the review,

as a reply we only ask for consideration to assign this manuscript as a review (not communication).

Thank you,

Jernej Pajek

Reviewer 2 Report

This review tries to indicate beneficial and adverse influence of PBDs on blood UA concentrations.

I felt that this manuscript was interesting, but the themes were not narrowed down enough. The introduction cannot explain the outline of the text. The conclusion is also obscure.

First, definition of PBD should be stated in detail in the introduction section. Are processed foods made from plants, such as noodles and soy milk, included in PBDs? Additionally, I was confused with differences between vegetable food and PBDs. (I mean Line 175 vs. Line 218.) Definition of exclusive PBDs, vegetarian diets, and vegan diets will be helpful to some readers.

Line 28: Abbreviations in the text should be independent of that in the abstract. That is, first appearance of abbreviations in the text should be spelled out again.

There is no explanation of the reason why the author focused uric acid in the introduction section. The authors cited influence of PBDs on chronic disease and/or lifestyle-related disease in the first paragraph (I cannot know differences between the 1st and 2nd sentences). Thus, relationship of UA with these diseases should be also indicated.

Contribution of diets to blood UA concentration is much less than that of genetic factors (see the reference by Major et al (2018)) including sex differences (Line 272-273). Additionally, most of uric acid is de novo synthesized in the body, and influence of exogenous purine on blood UA concentration is not so large (Line 337-339). None the less for the very low contribution of diets to blood UA concentrations, why should PBDs be discussed on blood UA levels? The authors should consider and explain relevance and benefits of reviewing such related articles in the introduction section, but not in the later section.

I cannot find priority of dietary purine in this manuscript. In the introduction section, the authors cited only purine contents, but not other nutrient factors, in foods (Line 40-47), implying purine contents as the main theme of this review. However, I cannot understand the reason why the authors focused on purine contents in foods. (On the other hand, in the main body of the text, the authors discussed various nutrient factors in a balanced manner.) Additionally, in the line of 54-55, the authors indicated some nutrient factors, including purine, that are involved in purine synthesis. However, the authors did not indicate contribution of purine to the UA synthesis and/or blood UA levels or did not compare contribution of purine with other nutrients.

In the conclusion section, the authors should clearly state their conclusion of relationship between uric acid and plant-based nutrition. What are the conclusions for healthy subjects and for gout and/or CKD patients?

Line 54: Nucleic acids are synthesized from amino acids in the body, independent of animal protein or plant protein. I felt that this sentence attempts to make animal protein bad.

Line 208-217: This paragraph may not be suite for this subheading.

Line 185: I cannot believe free saturated fatty acids in legumes. If so, please cite literatures that show no saturated fatty acids in legumes.

Table 2: The unit in the high content column should be ‘200-300 mg/100 g’.

Line 270-287: This section should be summarized.

Line 326-329: In addition to influence of PBDs on UA levels in chronic kidney disease (CKD) patients, these diets are considered to supply abundant potassium (Line 363-364), which may exacerbate symptoms of CKD. This view point should not be ignored and should be discussed or at least mentioned.

Line 335 (and maybe also Line 334): Table 1 should be Table 3. In the 3rd table, ‘WFPBD’ should be defined.

Line 375-376: ‘these diets’ should be shown as the exact terms. Is it true that elimination of plant foods potentially harmless to the health of people?

That’s all.

Author Response

Please see attachment also,

B. Jakše et al.

Uric acid and plant-based nutrition

Answers to reviewers

Dear editor, reviewers,

thank you for a thorough and concise effort in reviewing our manuscript. We have modified the manuscript and prepared a point-by-point answers to reviewer's comments, as evident below. We believe that the manuscript has been improved and hope that you will find the modified version satisfactory.

Answers to reviewer 1

This review tries to indicate beneficial and adverse influence of PBDs on blood UA concentrations. I felt that this manuscript was interesting, but the themes were not narrowed down enough. The introduction cannot explain the outline of the text. The conclusion is also obscure.

1. First, definition of PBD should be stated in detail in the introduction section. Are processed foods made from plants, such as noodles and soy milk, included in PBDs? Additionally, I was confused with differences between vegetable food and PBDs. (I mean Line 175 vs. Line 218.) Definition of exclusive PBDs, vegetarian diets, and vegan diets will be helpful to some readers.

1.1. ANSWER: We have added a new section at the beginning of the manuscript, immediately after introduction, to explain the variants of PBDs, as follows (Lines 54-86):

“Characterization of plant-based diets

The concept of a “plant-based diet” can have various definitions in the scientific literature; from excluding all animal source foods to including “only” a greater intake of vegetables, fruit, fruit juices, cereals, and legumes, while still including the intake of fish, pork, and yoghurt. Some scientific publications categorize PBDs by its actual content, e.g. semi-vegetarian (a typical western-type diet with a reduced frequency of consuming animal source foods), pesco-vegetarian (PBD including seafood with or without eggs and dairy products), and vegan diets (no animal source foods) (Williams and Patel, 2017). Unfortunately, the dichotomous division to vegetarian and non-vegetarian diet does not offer an overall insight into the quality of diet.

When we talk about health improving dietary interventions and therapeutic effects of the vegan diet, the term “vegan diet” may not be unanimously representative for the beneficial type of a diet. It may include too many highly processed and caloric foods, such as vegetable oils (sunflower, olive, pumpkin oil), exotic fats (coconut and palm fat, cocoa), processed cereals, highly processed packaged food, too much salt and added sugar. All these can adversely change the nutritional composition of the primarily beneficial PBD and, consequently, the effect on the body. The vegan cohort in the Oxford EPIC study for example, while consuming no animal-based foods, still consumed substantial amounts of saturated fat, and refined carbohydrates (Davey et al., 2003). A well-planned vegan diet should include a much larger intake of fruit, vegetables, nuts, and legumes in comparison with other non-vegetarian and vegetarian diets (Tantamango-Bartley et al., 2016). A “well-planned” refers to the intake of macronutrients and micronutrients, which should be adjusted according to the individual’s needs (Kahleova, Levin and Barnard, 2017).

Strict and exclusive vegan diet, often referred to as a PBD in a narrow sense, includes minimally processed fruit, vegetables, wholegrain cereals, legumes, nuts, seeds, herbs, and spices, while it excludes all animal source foods, such as various types of red meat, poultry, fish, eggs, and dairy products (Ostfeld, 2017). A whole food plant-based diet (WFPBD) is a variant of an exclusive PBD, described by Campbell and Campbell (2005). It is based on plant foods that are minimally processed. It includes little or no added fat and excludes refined carbohydrates. WFPBD allows for ad libitum intake of whole grains, fruits, vegetables, and legumes. In moderation, it also includes nuts, seeds, avocados, natural sweeteners, and certain soy or wheat products that do not contain added fat (e.g., tofu). Heavily processed foods, on the other hand, are not included in a WFPB diet. This means avoiding highly refined grain products (e.g., white rice, white flour), foods containing added sugars or artificial sweeteners (e.g., sugar, high fructose corn syrup), and foods containing added fat or refined fat (e.g., olive oil, coconut and palm oil) (Campbell and Jacobson, 2013).”

Also, we appreciate the comment about the unclear distinction between the vegetable foods and PBDs. Therefore we removed the misleading section heading (“Vegetable foods and serum UA concentration”) and attached the part of the text concerning legumes to the “Dietary factors and risk for hyperuricemia” section. We believe that the organization of discussion is clearer now.

2. Line 28: Abbreviations in the text should be independent of that in the abstract. That is, first appearance of abbreviations in the text should be spelled out again.

2.1. ANSWER: Yes, this was corrected to: “Plant-based diets (PBDs)«.

3. There is no explanation of the reason why the author focused uric acid in the introduction section. The authors cited influence of PBDs on chronic disease and/or lifestyle-related disease in the first paragraph (I cannot know differences between the 1st and 2nd sentences). Thus, relationship of UA with these diseases should be also indicated.

3.1. ANSWER: We wanted to stress that while PBDs are beneficial for many key chronic non-communicable diseases in developed countries, there is a concern about the purine content and uric-acid generation potential of these diets. So we have added this explanation at the end of the first paragraph of the introduction:

While this solid and consistent evidence from interventional and observational studies support the adoption of PBDs for key non-communicable chronic diseases in developed countries, we observe a common concern between patients and health-care practitioners about the intake of many otherwise health-promoting plant foods (such as legumes, broccoli, spinach, etc.) because of their purine content and uric acid generation potential.”

3.2. ANSWER: the difference between the 1st and 2nd sentence is that the 1st sentence describes the studies of association of PBDs with the reduced risk, while 2nd sentence describes the impact of PBD in interventional studies to prevent and treat some key chronic diet-related diseases.

3.3. ANSWER: The exact relationship of UA with some key chronic diseases is explained in “Uric acid and health outcomes” section, 3rd paragraph:

Hyperuricemia is a main risk factor for gout but is also associated with chronic kidney disease, cardiovascular diseases, type 2 diabetes, and dyslipidemia (De Leeuw et al., 2002; Obermayr et al., 2008; Lv et al., 2013; Qiu et al., 2013; Soltani et al., 2013; Braga et al., 2016; Muiesan et al., 2016b, 2016a). Hyperuricemia is often a consequence of relative renal underexcretion of UA since more than 70% of its excretion is performed by the kidneys (Lipkowitz, 2012). It presents a risk factor for incidence and a more rapid progression of chronic kidney disease, and some studies have shown a reduction in chronic kidney disease progression with UA-lowering treatments (Levy and Cheetham, 2015)

4. Contribution of diets to blood UA concentration is much less than that of genetic factors (see the reference by Major et al (2018)) including sex differences (Line 272-273). Additionally, most of uric acid is de novo synthesized in the body, and influence of exogenous purine on blood UA concentration is not so large (Line 337-339). None the less for the very low contribution of diets to blood UA concentrations, why should PBDs be discussed on blood UA levels? The authors should consider and explain relevance and benefits of reviewing such related articles in the introduction section, but not in the later section.

I cannot find priority of dietary purine in this manuscript. In the introduction section, the authors cited only purine contents, but not other nutrient factors, in foods (Line 40-47), implying purine contents as the main theme of this review. However, I cannot understand the reason why the authors focused on purine contents in foods. (On the other hand, in the main body of the text, the authors discussed various nutrient factors in a balanced manner.) Additionally, in the line of 54-55, the authors indicated some nutrient factors, including purine, that are involved in purine synthesis. However, the authors did not indicate contribution of purine to the UA synthesis and/or blood UA levels or did not compare contribution of purine with other nutrients.

4.1. ANSWER: Although the reference of Major et al (2018) does indeed conclude that in contrast with genetic contributions diet explains very little variation in serum urate levels, this study is seriously limited by the methodology of diet estimation – they aggregated data from ARIC, CARDIA, CHS, FHS and NHANES III cohorts with significantly different diet recall questionnaires (with all their limitations), they used techniques such as combining of questionnaire items, data aggregation to calculate diet pattern related scores and all this on a cross-sectional samples without estimation of portion sizes. The significant impact of purine content in foods on serum UA levels is nicely reviewed in Choi et al, Ann Intern Med, 2005. We however do accept the above-mentioned concerns, so we have rewritten the second paragraph of the introduction to give the following explanation on rationale of focusing on PBDs and purine content:

Although the serum UA concentration in each individual represents a complex interplay between non-modifiable factors, (e.g. genetics), and modifiable factors, (e.g. body weight and lifestyle), the diet and its purine content do play a part in this. In the literature, there is a lack of data evaluating the impact of exclusive PBDs on serum UA, especially from interventional studies, which did not control for this parameter in most cases. In this review, we will focus on the impact of some important individual dietary contents and foods but also on the collective impact of various PBD variants on serum UA levels. Purines may originate from body’s endogenous synthesis or be ingested with foods of variable purine content. Since UA is the end-product of metabolism of purines, there is a concern about increased serum UA concentrations with the transition from a Western-type mixed diet to an exclusive PBD due to introduction of the regular intake of plant foods with a high purine content, containing more than 200–300 mg purine per 100 g of the product. We will present and summarize the data on serum UA level changes with transition from mixed western-type diet to exclusive PBDs. Additionally, we review other potentially modifiable diet-related factors that can affect UA levels, such as type of purines ingested and the role of other phytonutrients. Finally, we will describe some modifications of exclusive PBDs that patients with UA associated health problems could follow.”

Also, please note, that we have elaborated the relative minor effect of diets on the serum uric acid concentration in “Purine content of PBD lifestyle may not be an unavoidable challenge” section, 3rd paragraph:

There are additional factors besides purine content of PBD, which determine the individual UA level response to PBD transition. First, Messina et al. (2011) suggest that approximately two-thirds of the total amount of urate in the body is endogenously synthesized and that “only” one-third comes from dietary purines. The production of serum UA depends on the balance between purine ingestion, de-novo synthesis in cells, recycling, and the degradation function of xanthine oxidase at the distal end of the purine pathway (Maiuolo et al., 2016). The association between a purine-rich diet, increased plasma UA and gout has long been recognized but a purine-rich diet may be responsible for only 59-118 µmol/L of serum UA (Ekpenyong and Akpan, 2014). The impact of dietary purines on plasma UA concentration is further modified by the frequency of consumption, cooking methods, bioavailability, types of purines, the presence of other plant-based food components, and lifestyle factors (Zgaga et al., 2012; Kaneko et al., 2014). So dietary purine content alone does not represent the major impact on serum UA concentration.”

5. In the conclusion section, the authors should clearly state their conclusion of relationship between uric acid and plant-based nutrition. What are the conclusions for healthy subjects and for gout and/or CKD patients?

 5.1. ANSWER: Regarding the relationship between uric acid and plant-based nutrition, and the distinction between healthy and gout/CKD patients, we have stated the following in the conclusion section:

As regards the relationship between the cumulative impact of plant-based nutrition and serum UA level, studies that compared UA serum concentrations in vegetarians and non-vegetarians have consistently shown a lower mean UA serum concentration in vegetarians. The concern about increased serum UA concentrations in the transition from a Western-type mixed diet to an exclusive PBD from whole food sources, with the regular intake of high-purine PB foods, is not needed in healthy normouricemic individuals since our results show, that the increase is minimal and well within the normal range. Especially females on such a long-term diet could even further increase the intake of purine rich plant foods to be able to reach a putative optimal serum UA concentration range. On the other hand, high-purine plant-based foods should be eaten with caution or replaced with lower purine content alternatives if a patient already has advanced kidney disease or gout. In these patients, UA lowering pharmacological agents remain the mainstay of therapeutic approach. We need more data about the impact of exclusive PBD in this patient population, however lower purine content plant foods are readily available and may be included in a healthy and nutritionally adequate plant-based nutrition plan, an example of which we presented in this review.«

6. Line 54: Nucleic acids are synthesized from amino acids in the body, independent of animal protein or plant protein. I felt that this sentence attempts to make animal protein bad.

 6.1. ANSWER: we had no intention to make animal protein look bad. We have rewritten this sentence to yield:

UA is the end product of purine metabolism, largely derived from endogenous synthesis but in minor part also from exogenous sources such as foods with a high purine content, alcohol and fructose drinks.”

7. Line 208-217: This paragraph may not be suite for this subheading.

7.1. ANSWER: Yes, we have moved this paragraph to follow the meal plan modification table 3, where it is more suitable.

8. Line 185: I cannot believe free saturated fatty acids in legumes. If so, please cite literatures that show no saturated fatty acids in legumes.

8.1. ANSWER: Legumes (called also »pulses«) such as kidney beans, cannellini beans, Great Northern beans, navy beans, fava beans, cranberry beans, black beans, pinto beans, soy beans, black-eyed peas, chickpeas, and lentils, are naturally low in fat, are practically free of saturated fat, and because they are plant foods, they are cholesterol free as well. One serving of legumes, which is one-half cup (118 g), provides about 115 calories, 20 g of carbohydrate, 7–9 g of fiber, 8 g of protein, and 1 g of fat. Please see the cited reference: Polak, Phillips and Campbell, 2015: https://www.ncbi.nlm.nih.gov/pmc/articles/PMC4608274/

(It is open source, so you may check easily). Also, USDA database on nutrients may be consulted – there is 0,00g saturated fat per 100g of most frequently used legumes: https://ndb.nal.usda.gov/ndb/

 9. Table 2: The unit in the high content column should be ‘200-300 mg/100 g’.

 9.1. ANSWER: Yes, this was corrected.

10. Line 270-287: This section should be summarized.

10.1. ANSWER: This is one of the few available prospective interventional data on what happens with serum uric acid in transition from western-type mixed diet to an exclusive whole food PBD. So we propose that the concise description of the methods used and results obtained remains in this manuscript.

11. Line 326-329: In addition to influence of PBDs on UA levels in chronic kidney disease (CKD) patients, these diets are considered to supply abundant potassium (Line 363-364), which may exacerbate symptoms of CKD. This view point should not be ignored and should be discussed or at least mentioned.

11.1. ANSWER: Yes, we agree. So the warning about the need to monitor serum potassium levels was included:

“Also, due to a larger potassium food content, serum potassium levels need to be monitored as well - we recommend additional control in seven and then 21 days after transition to a PBD in chronic kidney disease patients grade 3b, 4 and 5.«

12. Line 335 (and maybe also Line 334): Table 1 should be Table 3. In the 3rd table, ‘WFPBD’ should be defined.

12.1. ANSWER: Yes, this was all corrected.

13. Line 375-376: ‘these diets’ should be shown as the exact terms. Is it true that elimination of plant foods potentially harmless to the health of people?

 13.1. ANSWER: We apologize for the mistake, harmless should be replaced with harmful.

That’s all.

Submission Date

10 July 2019

Date of this review

16 Jul 2019 07:55:23

Date of answer to reviewers: 20/7/2019
